# Upcycling Grape Pomace in a Plant-Based Yogurt Alternative: Starter Selection, Phenolic Profiling, and Antioxidant Efficacy on Human Keratinocytes

**DOI:** 10.3390/foods14244294

**Published:** 2025-12-13

**Authors:** Andrea Torreggiani, Mario Caponio, Daniela Pinto, Giorgia Mondadori, Vito Verardo, Ana María Gómez-Caravaca, Michela Verni, Carlo Giuseppe Rizzello

**Affiliations:** 1Department of Soil, Plant, and Food Sciences, University of Bari, 70125 Bari, Italy; andrea.torreggiani@uniba.it (A.T.); mario.caponio@uniba.it (M.C.); 2Department of Environmental Biology, “Sapienza” University of Rome, 00185 Rome, Italy; carlogiuseppe.rizzello@uniroma1.it; 3Human Microbiome Advanced Project (HMAP), Giuliani S.p.A, 20129 Milan, Italy; dpinto@giulianipharma.com (D.P.); gmondadori@giulianipharma.com (G.M.); 4Institute of Nutrition and Food Technology ‘José Mataix’, Biomedical Research Center, University of Granada, Avda del Conocimiento s/n, Armilla, 18100 Granada, Spain; vitoverardo@ugr.es; 5Department of Nutrition and Food Science, University of Granada, Campus of Cartuja s/n, 18071 Granada, Spain; anagomez@ugr.es; 6Research and Development of Functional Food Center (CIDAF), Avda. del Conocimiento, 37, 18016 Granada, Spain; 7Department of Analytical Chemistry, University of Granada, Campus of Fuentenueva s/n, 18071 Granada, Spain

**Keywords:** grape pomace, plant-based gurts, lactic acid bacteria, phenolic compounds, oxidative stress, polyphenols biotransformation, gurt fermentation

## Abstract

Due to its appealing composition, grape pomace (GP), the major by-product of the wine industry, could be considered an ideal candidate for innovative functional foods development. In this study, a rice/GP-yogurt alternative, also known as gurt, fermented with selected lactic acid bacteria, was designed. An extensive characterization of the gurts led to the selection of the one fermented with *Lactiplantibacillus plantarum* T0A10. The strains showed good pro-technological performances (fast acidification and growth up to 9 log cfu/g in the specific plant-based composite substrate), as well as the ability to increase DPPH radical scavenging activity compared to the unfermented control (57% against 40%). Then, an in-depth focus on the effect of fermentation on phenolic compounds and their related antioxidant efficacy on human keratinocytes was provided, elucidating a compound/function relationship. Fermentation significantly modified the phenolic profile of the gurt, reducing glycosylated forms of flavonols and phenolic acids and increasing the content of catechin and pyrogallol (more than 100 mg/kg combined). Such modification was responsible for significantly up-regulating (*p* < 0.05) the expression of the antioxidant enzyme superoxide dismutase 2, thus protecting NCTC 2544 cells against oxidative stress. Overall, these findings provide a foundation for developing value-added products from GP, supporting both circular economy initiatives and functional ingredient innovation.

## 1. Introduction

The transition of food systems towards a circular model has resulted in a heightened level of scientific and industrial interest in the valorization of by-products that were previously regarded as waste [1]. Among these, grape pomace, the heterogeneous mixture of skins, seeds, and residual stems left after pressing, is of particular interest due to both its scale of generation and its unusually rich reservoir of bioactive constituents [2]. As a by-product of the global wine industry, grape pomace (GP) is produced in millions of tons on an annual basis. Its disposal can incur environmental, logistical, and economic concerns if it is not strategically repurposed [1,2].

The intrinsic composition of GP, including dietary fiber, structural polysaccharides, residual proteins, organic acids, lipids, and notably a rich profile of polyphenols, positions it as a promising input for food, nutraceutical, and biomaterial applications [2,3]. In most cases, GP is either composted or used to extract compounds of industrial interest, such as organic acids. It can also be used for distillation, although this practice has significantly declined in recent years [4]. It should also be noted that, unlike other approaches (e.g., the extraction of compounds of interest), which generate more residues, those involving GP’s entire upcycle should be preferred. A recent review of the chemical diversity of GP phenolics (anthocyanins, flavan-3-ols and their procyanidin polymers, flavonols, phenolic acids, and stilbenes) outlined the importance of expanding application across functional foods, cosmetics, and health-oriented formulations [3].

Concurrently, propelled by several factors including lactose intolerance, dairy protein allergy, sustainability perceptions, and evolving culinary preferences, consumers’ demand for plant-based dairy analogous has surged [5]. In this category, spoonable yogurt-like fermented products, often referred to as “gurts” [6,7,8,9], have emerged as a technically challenging yet high-potential segment. A comparison of plant matrices with dairy yogurt reveals significant disparities in protein quality, buffering capacity, fermentable carbohydrate profiles, and the presence of antinutritional factors. These variables contribute to the complexity of acidification kinetics, texture formation, and flavor generation [10]. In this matter, the role of lactic acid bacteria (LAB), the selection of the ingredients and the optimization of the production process, and their structure–function relationship during fermentation, represent an opportunity to improve both sensory and nutritional properties of gurts [10].

Recent reviews on GP valorization document product categories and health-relevant endpoints but often stop short of pairing high-resolution phenolic analytics with ex vivo bioactivity in a single study design [11]. Meanwhile, research on plant-based yogurt-like alternatives [5] emphasizes the importance of fermentation technology, texture, and sensory aspects, with relatively few examples that add a complex polyphenol-rich by-product and then interrogate both its chemical fate and biological relevance after fermentation [5]. The present work aims to address this gap.

Hence, in this study, a fermented plant-based gurt enriched with grape pomace was developed. Integrating GP into gurts combines two trends: circular bioeconomy valorization and the nutritional/functional upskilling of plant-based fermented foods. GP acts as a source of soluble and insoluble fibers and supplies a complex of phenolic compounds whose antioxidant and biological effects could elevate the functional profile of the finished product [2].

In light of this, a two-stage investigation was designed aiming at (i) screening LAB-fermented plant-based gurt prototypes containing grape pomace based on suitable biochemical, nutritional, and functional quality; and (ii) selecting the most promising beverage prototypes for in-depth phenolic profiling and antioxidant testing on human keratinocytes.

## 2. Materials and Methods

### 2.1. Raw Materials and Microorganisms

The grape pomace (*Vitis vinifera* L. cultivar Primitivo) used in this work, provided by a winery in Southern Apulia (Carmiano, Italy) following a seven-day maceration phase, was dried at 65 °C for 60 min in a ventilated oven (Argolab, Carpi, Italy). Then GP was finely ground with a laboratory mill Ika-Werke M20 (GMBH and Co. KG. Staufen, Germany) to obtain a powder further sieved with a 150 μm mesh. The GP proximate composition was the following: proteins, 11%; lipids, 7%; ashes, 10%; carbohydrates, 17%; and total dietary fiber, 42%.

Commercial rice flour (Bioalimenta S.r.l., Chieti, Italy) was also used. Its proximal composition (g/100 g on dry matter) was as follows: total carbohydrates, 72.00; total dietary fibers, 3.20; lipids, 2.50; proteins, 8.20; and ashes, 0.61.

Three lactic acid bacteria strains, previously used for the fermentation of plant-based yogurt alternatives, *Lacticaseibacillus rhamnosus* SP1 [12] and *Leuconostoc pseudomesenteroides* DMS 20193 [7], or selected for the best pro-technological performance during fermentation of grape pomace-based matrices, *Lactiplantibacillus plantarum* T0A10 [13], were used as starters. LAB strains were singly cultivated in De Man, Rogosa, and Sharpe (MRS Oxoid, Basingstoke, Hampshire, UK) at 30 °C until the late exponential phase of growth was reached (ca. 16 h). Before the inoculum, cells were harvested by centrifugation (10,000× *g* for 10 min at 4 °C), washed twice in sterile 50 mM phosphate buffer pH 7.0, and resuspended in tap water.

### 2.2. Gurts Production Process

For the gurt making, rice and GP flours were resuspended in tap water at 15 and 5%, respectively. The pH of the mixture was adjusted to approximately pH 6.0 using food-grade E500 sodium bicarbonate (Solvay, Brussels, Belgium), and the mixture was gelatinized at 80 °C for 15 min. Gurts were then cooled at 30 °C prior to the inoculum of the starters. Strains (*Lp. plantarum* T0A10, *Lc. rhamnosus* SP1, and *Leuc. pseudomesenteroides* DMS20193) were singly inoculated at a cell density of approx. 7.0 log cfu/g, obtaining G-T0A10, G-SP1, and G-DSM, respectively. Fermentation was carried out at 30 °C for 18 h. An uninoculated control (G-Ct) was also prepared for each thesis and incubated in the same conditions. After fermentation, gurts were refrigerated at 4 °C for 14 days.

### 2.3. Microbiological and Biochemical Characterization

LAB were determined on MRS agar medium (Oxoid), supplemented with cycloheximide (0.1 g/L), and the plates were incubated in anaerobic conditions at 30 °C for 48 h.

The proximate composition of the gurts (proteins, lipids, moisture, total dietary fiber, and ash) was determined according to the Approved Methods of the American Association of Cereal Chemists 46–11.02, 30–10.01, 44–01.01, 32–05.01, and 08–01.01 [14].

The pH of the gurts was monitored with a FiveEasy Plus pH meter (Mettler-Toledo, Columbus, OH, USA) over a period of 18 h, and analyses were conducted at 2 h intervals. The kinetics of acidification were modeled using the Gompertz equation as modified by Zwietering et al. [15]:$$y=k+A\;exp\{-exp[(V_{max}\;e/A)(-t)+1]\}$$
where *y* is the acidification extent expressed as ΔpH at time *t* (pH/units/h), *k* is the initial level of the dependent variable to be modeled (pH units), *A* is the pH difference between inoculation and the stationary phase, *V_max_* is the maximum acidification rate, and *λ* is the length of the latency phase expressed in hours.

Total titratable acidity (TTA) was determined on 10 g of product homogenized with 90 mL of distilled water and expressed as a quantity (mL) of 0.1 M NaOH needed to reach a pH of 8.3.

For the quantification of organic acids and amino acids (FAA), water/salt-soluble extracts were prepared according to the method described by Weiss et al. [16]. More specifically, gurts containing 1 g of dry material were added to 4 mL of 50 mM Tris-HCl, pH 8.8, extracted for 1 h at 4 °C, and then centrifuged to obtain the extracts. The kits K-DLATE and K-ACET (Megazyme, Bray, Ireland) were used for the determination of lactic and acetic acid concentrations, following the manufacturer’s instructions. FAAs were determined using the Biochrom 30+ Amino Acid Analyzer (Biochrom Ltd., Cambridge Science Park, UK) with a Li cation exchange column (20 × 0.46 cm internal diameter [17].

The gurts’ total polyphenol concentration (TPC) was measured on methanolic extracts, prepared by treating the samples with 80% methanol at a ratio of 1:10 (1 h of stirring at 4 °C) and subsequently by centrifuging at 10,000 rpm for 20 min. TPC was determined using the Folin–Ciocalteu reagent (Sigma Chemical Co., Burlington, MA, USA) as described by Slinkard and Singleton [18] and expressed as gallic acid equivalent (GAE).

Antioxidant activity was determined by assessing the capacity to scavenge the DPPH (2,2-diphenyl-1-picrylhydrazyl) radical [19]. The scavenging activity was expressed as the percentage of scavenged DPPH compared to a blank containing no antioxidants, as follows: DPPH scavenging activity (%) = [(blank absorbance − sample absorbance)/blank absorbance] × 100 (after 10 min of reaction). The synthetic antioxidant butyl hydroxytoluene (BHT) was included in the analysis as a reference (75 ppm).

Aiming at investigating the gurts’ microbial and biochemical stability, LAB cell density, pH, TTA, organic acids, and FAA concentration, as well as DPPH-radical scavenging activity, were analyzed, as described above, after 7 and 14 days of refrigerated storage.

### 2.4. Color and Sensory Analysis

The chromaticity coordinates of the gurts were obtained by a CS-10 colorimeter (CHN Spec Technology, Hangzhou, China) and reported as color difference, ${\Delta E}_{\mathrm{ab}}^{*}$, calculated by the following equation:$${\Delta E}_{\mathrm{ab}}^{*}=\sqrt{{\left({\Delta L}^{*}\right)}^{2}+{\left({\Delta a}^{*}\right)}^{2}+{\left({\Delta b}^{*}\right)}^{2}}$$
where ${\Delta L}^{*}$, ${\Delta a}^{*}$, and ${\Delta b}^{*}$ are the differences for $L^{*}$, $a^{*}$, and $b^{*}$ values between sample and reference (a white ceramic plate having $L^{*}$ = 92.2, $a^{*}$ = 0.15, and $b^{*}$ = 0.85).

Sensory analyses were carried out by 15 trained panelists (6 men and 9 women; average age: 31 years, range: 24–41 years) with demonstrated abilities and prior expertise in cereal-based product assessment. A two-hour training session was performed, and the assessors evaluated the descriptors to be included in the sessions (Appendix A). In particular, the following descriptors were chosen: color intensity (Cl), uniformity (Uf), adherence to spoon (Ad), and presence of particles for the appearance (Pr); overall odor intensity (Od), pungent smell (Pn.s), acidic smell (Ac.s) for the odor; sweet (Sw.t), salty (St.t), bitter (Bt.t), acidic (Ac.t) and astringent (As.t) for the taste; sweet (Sw.at), astringent (As.at), and earthy (Er.at) for the aftertaste evaluation.

Sensory attributes were scored on a scale from 0 to 10, with 10 being the highest score. Sensory evaluations were carried out following the independent method of the “Sensory analysis—Methodology—Flavour Profile” methods (ISO 6564-1985 [20]) with some modifications. In detail, the library of the Environmental Biology Department of the Sapienza University of Rome (Italy) was used instead of cabinets as previously proposed by Elia [21]. Enrolled panelists, who did not suffer from any food intolerances or allergies, received information on the objectives of this study and provided written informed consent. Three separate sessions were conducted, where gurts were served in a randomized order and encoded with three-digit random numbers. A glass of water was drunk by the panelists alongside tasting the samples.

### 2.5. Study of Gurts Phenolic Profile

#### 2.5.1. Extraction of Phenolic Compounds

Aiming at studying the modifications that occurred during fermentation and their impact on in vitro antioxidant properties, G-Ct and G-T0A10 were selected for further characterization. Phenolic compounds were extracted using an ultrasound bath, and a mixture of ethanol/water 80/20 (*v*/*v*) was used as extractant solvent [22]. More specifically, 2 g of the sample was added to 40 mL of solvent and extracted twice in an ultrasonic bath at 40 °C. Then, extracts were subjected to vacuum evaporation at 40 °C with a rotavapor (R-100 Buchi, Barcellona, Spain) and then reconstituted with 2 mL of mixture methanol:water (50/50 *v*/*v*) for the analysis [22].

#### 2.5.2. Qualitative and Quantitative Analysis of Phenolic Compounds

The separation and identification of phenolic compounds was carried out with a BRUKER HPLC (Elute PLUS LC series from Bruker) coupled to a micro Q/TOF mass spectrometer with mass detector COMPACT from Bruker (Bruker Daltonics GmbH & Co., KG—Bremen, Germany). Ionization was carried out by an electrospray ionization (ESI) source operating in negative mode under the same conditions reported by Díaz-de-Cerio et al. [23]. Compounds were separated using a Poroshell 120 EC-C18 (4.6 mm × 100 mm, particle size 2.7 µm) (Agilent Technologies, Santa Clara, CA, USA). The gradient elution was carried out using water containing 1% acetic acid as solvent system A and acetonitrile as solvent system B, and applied as follows: 0 min, 2.5% B; 8 min, 10% B; 12 min, 15% B; 13 min, 16% B; 15 min, 18% B; 18 min, 20% B; 22 min, 100% B; 28 min, 2.5% B. The sample volume injected was 3 µL, and the flow rate used was 0.8 mL/min. Data integration and processing were performed using the Data Analysis 4.2 software. For the quantification of phenolic compounds, standard solutions of quercetin, rutin, apigenin 7 glucoside, and resveratrol solubilized in methanol; vanillic acid, chlorogenic acid, and catechin solubilized in methanol/water (50/50 *v*/*v*); and gallic acid solubilized in distilled water were used. The concentration of phenolic compounds used as the standard varied from 0.4 to 65 ppm, and R^2^ values of the regression equations we all higher than 0.99.

#### 2.5.3. Quantification of Proanthocyanidins

Proanthocyanidins were analyzed by HPLC-FLD, performed by an Agilent 1200 Series (Agilent Technologies), equipped with a Luna Hilic column (150 × 2.0 mm; 3 µm) and a fluorescence detector with an excitation wavelength of 290 nm and an emission wavelength of 325 nm. The separation was performed with 98% acetonitrile and 2% acetic acid (A) and 95% methanol, 3% water, and 2% acetic acid (B) as described by [24].

### 2.6. Study of Gurts Antioxidant Potential on Human Keratinocytes

#### 2.6.1. Human Keratinocytes Cell Culture

Human keratinocyte NCTC 2544 cells (HL97002) were supplied by Istituto Nazionale di Ricerca sul Cancro (Milano, Italy).

The immortalized line of human keratinocytes NCTC 2544 was cultured in RPMI 1640 medium, supplemented with 10% fetal bovine serum (FBS), 2 mM L-glutamine, penicillin (100 U/mL)/streptomycin (100 U/mL), and 50 mg/mL gentamicin. NCTC 2544 cells were maintained in 25 cm^2^ sterile culture flasks, incubated at 37 °C in a humidified atmosphere with 5% CO_2_.

#### 2.6.2. Proliferation Assay

The MTT assay was carried out according to the method by Mosmann [25] with slight modifications. Briefly, NCTC 2544 were seeded in 96-well plates, at a density of 5 × 10^4^ cells/well and incubated at 37 °C-5% di CO_2_ until reaching approximately 80% confluency. Subsequently, the medium was removed from the wells, and the cells were incubated with the fermented and unfermented samples diluted in RPMI 1640 medium, at the following concentrations: 0.1 mg/mL, 0.05 mg/mL, and 0.01 mg/mL. Each sample was tested in triplicate, and untreated cells were used as a control. Cells were then incubated for 24 h at 37 °C-5% CO_2_. After the incubation period, the medium was removed, and 100 µL of MTT reagent was applied to the cells (stock solution 5 mg/mL in DPBS, diluted 1:10 in RPMI 1640 medium without phenol-red). The plate was incubated for 3 h in the dark at 37 °C-5% CO_2_. After incubation time, 100 µL of dimethyl sulphoxide (DMSO) was added to dissolve purple formazan product. The solution was shaken in the dark for 15 min at room temperature. The absorbance of the solutions was read at 570 nm in a microplate reader (BioTek Instruments Inc., Bad Friedrichshall, Germany) (and at 630 nm as wavelength reference). Data were expressed as cell viability percentage, compared to the control cells, as per the following formula: % cell viability/ctrl = (Abs sample/Abs ctrl) × 100.

#### 2.6.3. Oxidative Stress Assay

NCTC 2544 were seeded in 96-well plates, at a density of 5 × 10^4^ cells/well, and incubated at 37 °C-5% CO_2_ until reaching approximately 80% confluency. Subsequently, the medium was removed from the wells and the cells were incubated with the fermented and unfermented samples diluted in RPMI 1640 medium, at the following concentration: 0.1 mg/mL for 24 h at 37 °C-5% CO_2_. After the incubation period, the medium was removed and substituted with 50 mM H_2_O_2_ diluted in RPMI 1640 medium, then further incubated for 2 h to induce oxidative stress. Untreated cells were used as a negative control (not stressed cells), and cells treated only with 50 mM of H_2_O_2_ were used as a positive control (stressed cells). Finally, the medium was removed, and the residual viability of the cells was measured using the MTT method as described previously. Each condition was tested in triplicate.

#### 2.6.4. qRT-PCR Assay

Gene expression of Superoxide Dismutase 2 (*SOD2*) in the NCTC 2544 cell line was evaluated through quantitative reverse transcription–polymerase chain reaction (qRT-PCR). The cells were seeded in 24-well plates at a density of 5 × 10^5^ cells/well and incubated at 37 °C-5% CO_2_ until reaching approximately 80% confluency; the medium was removed, and the cells were incubated for 24 h with fermented and unfermented samples at the following concentration: 0.05 mg/mL in RPMI 1640 supplemented medium. Untreated cells were used as a control. At the end of treatment, total RNA was extracted from NCTC 2544 according to the method described by Chomczynski and Mackey [26]. Complementary DNA (cDNA) was synthesized by reverse transcriptase using the commercial kit “PrimeScript^TM^ RT Reagent Kit (perfect Real Time)” (Takara Bio Inc., Kusatsu, Japan). The produced amplicons were quantified through the monitoring of the fluorescence emitted during the reaction by using the TaqMan^®^ (Applied Biosystems, Waltham, MA, USA) probe system. The following TaqMan^TM^ probes were used: Hs00167309_m1 (*SOD2*) and Hs99999905_m1 (GAPDH) as housekeeping gene. The obtained data were analyzed according to the 2^−ΔΔCt^ method [27].

### 2.7. Statistical Analysis

Analyses were carried out on samples obtained in three separate replicates, and each sample was analyzed in duplicate. Data were subjected to one-way ANOVA; paired comparison of treatment means was achieved by Tukey’s procedure at *p* < 0.05, using the software Statistica 12.5 (StatSoft Inc., Tulsa, OK, USA). Data from the in vitro analysis on the NCTC 2544 cell line were subjected to Student’s *t*-test, followed by Welch’s correction using the statistical software, GraphPad Prism 10.6 (GraphPad Software, San Diego, CA, USA, www.graphpad.com).

## 3. Results

### 3.1. Starter Selection

#### 3.1.1. Growth and Acidification

Aiming at selecting the most suitable starter among those used for gurt fermentation, the gurts’ main pretechnological features were evaluated. Cell density increases of almost 2 log cycles were observed, with values reaching, on average, 8.8 log cfu/g (Table 1).

All the strains were able to ferment and acidify the gurts, but with different intensities. On average, values of 4.55 and 4.06 mL were reached, respectively, for pH and TTA (Table 1), but the acidification kinetics highlighted different behaviors. *Lc. rhamnosus* SP1 had the lowest latency phase (λ) but was also the strain that determined the lowest acidification, with ΔpH between the inoculum and stationary phase (*A*) of 1.21 units (Figure 1). Whereas *Leuc. pseudomesenteroides* DSM20193 and *Lp. plantarum* T0A10, which showed similar acidification in terms of *A* and λ, were the ones with the lowest (0.094 ΔpH/h) and highest (0.186 ΔpH/h) *V_max_*, respectively.

#### 3.1.2. Organic Acids and Free Amino Acids

The significantly (*p* < 0.05) lower pH and higher TTA found, especially in the fermented gurts compared to G-Ct, were consistent with the production of organic acids. On average, 14.7 and 0.88 mmol/kg of lactic and acetic acids, respectively, were found in the fermented gurts (Table 1). G-DSM and G-T0A10 had the highest content of lactic acid, whereas in G-SP1 and G-DSM, the lowest concentration of acetic acid was found.

Before fermentation, total free amino acids (TFAA) were roughly 400 mg/kg, and after fermentation, decreases between 28 and 60% were observed in all the gurts except G-SP1 (Table 1). In G-Ct, Asp, Glu, Gly, Val, Arg, and Pro were the most abundant amino acids (25 ± 0.99, 53 ± 3.18, 27 ± 1.90, 36 ± 0.72, 35 ± 1.39, 66 ± 2.17 mg/kg, respectively). After fermentation, a significant reduction, less marked for G-SP1, in the concentration of Thr, Ser, Glu, Gly, Met, Ile, Leu, Tyr, and Phe was observed. When *Lp. plantarum* T0A10 was used for the fermentation, the concentration of GABA, Orn, His, and Arg almost doubled compared to G-Ct (Figure 2).

#### 3.1.3. Nutritional, Functional, and Sensory Characteristics

As expected, fermentation did not result in any significant changes in macronutrient content. Indeed, all gurts had the same composition: moisture, 75.4 ± 3.8%; proteins, 1.8 ± 0.07%; lipids, 0.8 ± 0.01%; ashes, 1.2 ± 0.18%; carbohydrates, 11.8 ± 0.57 of which total dietary fiber, 3.1 ± 0.16%.

Nevertheless, total phenolic content was affected by fermentation, which increased from 6 to 11% (*p* < 0.05) in G-SP1 and G-T0A10, respectively. Concurrently, compared to G-Ct, DPPH-radical scavenging activity increased up to 59% in the fermented gurts (Table 1).

Fermentation also caused changes in the gurts’ sensory profile, significantly affecting flavor and taste attributes; indeed, a clear separation between G-Ct and the fermented gurts was observed (Figure 3).

G-Ct had a different texture, with more perceived presence of particles, as well as a lesser adherence to the spoon. In G-Ct were also detected more earthy and astringents notes in the aftertaste. On the other hand, G-DSM, G-SP1, and G-T0A10 were characterized by a pungent odor and a more intense acid taste compared to G-Ct. G-Ct was also characterized by a paler and less vivid color compared to the fermented gurts. Indeed, when chromaticity coordinates were determined, samples significantly differed (*p* < 0.05) only for the *a** value (the red index), which was the highest in the fermented gurts, especially G-T0A10 and G-DSM (Table 2). Overall, the general acceptance was positive for fermented gurts, especially G-T0A10, with scores up to 7.4 ± 0.5, whereas G-Ct acceptance was scored 3.3 ± 0.8 points.

#### 3.1.4. Shelf-Life Monitoring

The main microbial, biochemical, and nutritional features of the gurts were also monitored during 14-day refrigerated storage (Table 1). In the fermented gurts, LAB cell density remained stable between 8.2 and 9 log cfu/g, whereas a slight but significant increase, up to 3 log cfu/g, was observed in G-Ct at the end of storage. Acidification slowly continued (*p* < 0.05) during refrigeration, leading to a further pH decrease and TTA increase by day 14, in all samples. Similarly, TFAA and TPC increased during storage, especially in G-T0A10, also leading to a significant increase in DPPH-radical scavenging activity, up to 78% (Table 1). Hence, G-T0A10 was selected for a more in-depth characterization of the phenolic profile and antioxidant activity on human keratinocytes.

### 3.2. Phenolic Compounds Profile

Phenolic compounds were separated and identified by RP-HPLC-ESI-Q/TOF. Appendix A summarizes the information related to the compounds tentatively identified: retention times (RT, min), experimental and calculated *m*/*z*, molecular formula, and error (ppm). A total of thirty-nine phenolic compounds, previously found in grape pomace or grapes [28,29,30,31,32,33,34,35,36,37,38,39,40,41,42], were identified.

Before fermentation, the most abundant compound was catechin, followed by quercetin glucuronide. Phenolic acids and their glucosides, as well as flavonoids and tannins, were also found (Table 3). Fermentation significantly modified the phenolic profile of the gurt. Indeed, syringol, as well as protocatechuic, caftaric, fertaric, and *p*-coumaric acids, were not found in G-T0A10, whereas pyrogallol was only found after fermentation. Although a lower number of compounds was identified in G-T0A10, the total amount was slightly higher (*p* > 0.05) in G-T0A10 compared to G-Ct.

Fermentation also affected proanthocyanidins, which were extracted and analyzed by HPLC-FLD (Figure 4). Polymeric forms higher than 6 units were below the detection limit for both samples, and only those with a degree of polymerization below 5 were found. No differences (*p* > 0.05) were observed between G-Ct and G-T0A10, for monomeric forms; on the contrary, dimers, trimers, tetramers, and pentamers were significantly lower in G-T0A10.

### 3.3. Proliferative Activity and Protection from Oxidative Stress

The MTT assay was employed to evaluate the viability of the NCTC 2544 cell line when exposed to varying concentrations of gurts. Overall, a tendency in increasing cell viability was observed after 24 h of exposure to low concentrations (0.1, 0.05, and 0.01 mg/mL) of G-Ct and G-T0A10, although data were not statistically significant compared to untreated cells.

When oxidative stress was induced with H_2_O_2_, a cytotoxic effect in NCTC 2544 cells was observed. Indeed, H_2_O_2_ reduced cell viability from 100 to 10.95%, whereas G-Ct and G-T0A10 counteracted the negative effect. In fact, their residual viability in treated cells was 22.71% and 24.10% for G-Ct and G-T0A10, respectively, although without significant difference among each other. Nevertheless, only G-T0A10 was able to significantly up-regulate the gene expression of superoxide dismutase 2 (Figure 5).

## 4. Discussion

Grape pomace is a robust source of dietary fiber and polyphenols with growing evidence for pharmacological properties, among which are antioxidant, antimicrobial, vasodilation, anti-inflammatory, antihyperglycemic, and antitumoral bioactivities [43]. Several studies have ascribed the wide range of bioactivity to phenolic compounds in grape pomace, and many have focused on enhancing extraction and analytical techniques. Still, for these advancements to be translated into food development contexts, more research is needed to broaden the potential applications of grape pomace, while promoting its valorization in a circular economy framework. Evidence shows successful incorporation in various products, including baked goods (bread, pizza, cookies, muffins), pasta, animal-based products (yogurt, raw and cooked meat), with technological advantages mostly related to the antioxidant activity, but also as an antimicrobial ingredient [11]. Nevertheless, specific applications in plant-based yogurt alternatives are lacking and require further investigation. The limited research on plant-based yogurt-alternative development with grape pomace opens up to significant opportunities, among which are (i) the optimization of grape pomace incorporation in gurts, (ii) the evaluation of the effect of fermentation on the bioavailability of grape pomace compounds in such plant-based matrices, (iii) the study of sensory features and consumer acceptance.

Based on the above considerations, this study aimed at investigating the potential of grape pomace to be used as a functional ingredient in a plant-based gurt containing rice and GP flour. The gurts’ formulation, as it was designed, had a relatively high amount of fibers, solely provided by GP, so that it could be defined as “high in fiber”. Indeed, according to the European Regulation (EC) No 1924/2006, the “high in fiber” nutritional claim can be used when fibers constitute more than 3 g/100 kcal of product, as is the case with the gurts developed in this study. The beneficial health effects of fiber in one’s diet are well known. They are responsible, in conjunction with polyphenols, for the prebiotic effect ascribed to GP [44,45]; indeed, multiple studies demonstrated that, thanks to its composition, GP can serve as a prebiotic ingredient fostering the growth of beneficial colonic microbiota [45,46].

In order to modify the structure of the rice/GP flour mixture and achieve a creamy consistency, a gelatinization treatment was performed prior to fermentation. Furthermore, given the potential interference of GP’s extremely acidic pH on both LAB growth and structure [13], the pH of the mixture was adjusted to 6 prior to the heat treatment.

Three lactic acid bacteria strains were screened based on their pro-technological features. Namely, *Leuc. pseudomesenteroides* DSM20193 was selected due to its ability to synthesize exopolysaccharides in cereal- and legume-based gurts [6,7], which are known for their prebiotic potential as well as the ability to act as hydrocolloids, thus improving the structure of the matrix [47]. *Lc. rhamnosus* SP1 is a commercial probiotic previously used in plant-based yogurt alternatives [12] and was selected to evaluate its aptitude as a probiotic starter. Finally, *Lp. plantarum* T0A10, isolated from quinoa and already used to ferment a GP-based sourdough [13], was selected for its antioxidant properties. Although with some differences, all the strains were well adapted to the matrix, growing almost 2 log cycles and acidifying the gurts. *Lc. rhamnosus* SP1 was the strain that showed the lowest growth and acidification. *Leuc. pseudomesenteroides* DMS20193 and *Lp. plantarum* T0A10 had a more pronounced acidification, suggesting a more efficient carbohydrate metabolism, which led to greater lactic acid accumulation, and in the case of *Lp. plantarum* T0A10, also acetic acid. Overall, values for pH, TTA, and lactic and acetic acids (Table 1) are in line with those previously reported in similar plant-based gurts [6,7,8].

Although fermentation led to an increase in some amino acids, especially in G-DSM and G-T0A10, TFAA decreased in all gurts compared to G-Ct (Table 1, Figure 2). The formulation, in fact, contained a low concentration of free amino acids, which is why LAB, being auxotrophic for many of these, use them for their own needs [48], explaining the reduction after fermentation. It can also be hypothesized that, as already observed in other plant-based gurts [6], the low protein concentration and the short fermentation time did not allow for the significant release of amino acids through proteolysis, commonly found in food matrices fermented with lactic acid bacteria.

Slight but significant increases in the total phenolic content were also observed in fermented gurts compared to G-Ct, confirming the well-known notion that LAB fermentation influenced polyphenol bio-accessibility, leading to higher antioxidant activity [49]. In addition to improving its nutritional and functional quality, fermentation also improved the gurts’ overall sensory acceptability, conferring a pleasant aroma and masking earthy and astringent notes often related to tannins. The sensory attributes differentiating the gurts, for color, aroma, taste, and texture, are in line with biochemical characterization (Table 1, Table 2 and Figure 3). Indeed, the increased acidity observed in the fermented gurts due to the production of lactic and acetic acids reflected the sensory attributes, which highlighted higher acidity notes for both smell and taste in G-DSM, S-SP1, and G-T010. Moreover, although fermentation reduced some of the earthy notes typically associated with grape pomace, the presence of residual astringent notes could be a potential formulation challenge and warrants further optimization.

The main microbial and biochemical features of the gurts were monitored during shelf-life. The robust LAB viability during storage and changes in the biochemical feature confirmed the suitability of all three strains for sustained fermentation stability under refrigerated conditions, as well as a constant metabolic activity, although reduced.

A fundamental challenge in the design of fortified food is to connect compositional changes with functional outcomes. Since *Lp. plantarum* T0A10 was the strain that better adapted to the matrix, demonstrating more balanced biochemical features, as well as the most appreciated organoleptic profile, G-T0A10 was selected for an in-depth characterization of the polyphenolic profile and antioxidant potential on human keratinocytes.

Although grape pomace represented just a small amount of the dry ingredients in the gurts, the phenolic compounds identified were provided exclusively by GP, rather than rice. Indeed, the white rice used in this study, unlike colored rice varieties or unmilled rice, has extremely low content of phenolic compounds [50], let alone compared to GP. All the compounds identified were previously identified in GP [28,29,30,31,32,33,34,35,36,37,38,39,40,41,42], which is known to be abundant in flavan-3-ols (catechin/epicatechin and procyanidins), anthocyanins (especially in red varieties), phenolic acids (gallic, caffeic), and flavonols (quercetin, kaempferol glycosides) [51].

In G-T0A10, fewer compounds were identified. A significant decrease in the concentration of phenolic acids and their glycosides, as well as glycosylated flavonols, was observed. On the contrary, flavanols, flavanones, and tannins increased. These compositional changes reflect the extensive microbial biotransformation of polyphenols during fermentation. The increase in flavan-3-ols (catechin, epicatechin gallate) and decrease in glycosylated flavonoids (Table 3) is consistent with bacterial glycosidase activity, which can liberate or convert bound phenolics into more bioactive forms [49]. The decline in proanthocyanidin oligomers indicates partial depolymerization into absorbable monomers and their further metabolization by lactic acid bacteria. Indeed, the detection of pyrogallol, which was not found in G-Ct, or protocatechuic acid, is consistent with the activity of tannin acyl hydrolase and decarboxylase on phenolic acids and gallotannins [52]. Overall, such modifications could improve antioxidant potential and bioavailability, reinforcing the role of fermentation as a tool to tailor functional properties of plant-based foods.

While indispensable for early screening, in vitro antioxidant assays should be complemented by ex vivo biological models to increase physiological plausibility and provide a pragmatic bridge toward function without the cost and complexity of clinical trials [53]. The conceptual and methodological gap between in vitro chemical antioxidant assays and in vivo outcomes has prompted a shift toward cell-based and ex vivo methods that incorporate biological membranes, enzymes, and endogenous antioxidants. Hence, aiming at investigating the biological relevance of the antioxidant potential of the grape pomace-based gurts, using a well-established human cell model responsive to oxidative stress, the ability of the fermented and unfermented gurts to protect NCTC 2544 cells against induced oxidative stress was evaluated through MTT assay. Keratinocytes are among the most widely used human cell lines for screening antioxidant and anti-inflammatory bioactivity of plant bioactive compounds [13,54,55]. Although the fermented gurts designed in this study are not intended for topical use, and further studies could focus on evaluating their gastrointestinal effects, phenolic compounds absorbed from foods can exert systemic antioxidant and cytoprotective effects, including in peripheral tissues such as skin [56,57].

Cells pretreated with G-Ct and G-T0A10 were able to counteract the negative effect of H_2_O_2_-induced stress, leading to higher residual viability compared to untreated cells. The gurts were also tested for their ability to regulate the expression of genes codifying for enzymes involved in oxidation reactions, such as *SOD2* (Figure 5). Superoxide dismutase 2 is an antioxidant enzyme that converts superoxide anion to hydrogen peroxide, playing a role in the protection of cells from oxidative stress [58]. Our experimental results showed that treating NCTC 2544 cells with GP-gurts up-regulated the production of this gene. This up-regulation was especially pronounced when cells were treated with fermented samples. The up-regulation indicates a shift towards a more efficient cell neutralization of superoxide, thus limiting oxidative damage to mitochondrial proteins, DNA, and lipids [59]. Moreover, it was demonstrated that *SOD2* overexpression enhances mitochondrial function and metabolic vasodilation, leading to improved cardiac function in mice [60], and can also regulate neuroinflammation by controlling the activity of microglial (primary immune cells of the central nervous system) [61].

The significantly higher *SOD2* up-regulation observed in cells pre-treated with G-T0A10, compared to G-Ct, lies in the fermentation process. Indeed, during fermentation, grape pomace polyphenols were metabolized into simpler, more bioavailable metabolites such as pyrogallol. Pyrogallol can act as a mild pro-oxidant, generating low levels of hydrogen peroxide that activate redox-sensitive transcription factors, particularly Nrf2, a protein essential in the defense against cells oxidative stress, thereby inducing antioxidant defense genes including *SOD2* [62,63]. Unlike non-fermented gurt, where phenolics remained in less reactive forms, the presence of pyrogallol likely provided a specific signal that enhances mitochondrial antioxidant responses through adaptive hormesis mechanisms. These findings suggest that the superior *SOD2* up-regulation observed with G-T0A10 is linked to the formation of pyrogallol during fermentation, rather than the total phenolic content. Of course, a synergic action of multiple phenolic compounds cannot be excluded. For instance, catechin and protocatechuic acid, which were found in higher concentrations in G-T0A10, can also activate the Nrf2 pathway, promoting the expression of antioxidant enzymes such as *SOD2* [63]. However, pyrogallol, a microbial metabolite of catechins and protocatechuic acid, is generally more effective at enhancing Nrf2-dependent gene expression due to its simpler structure and higher redox potential [63].

## 5. Conclusions

This study demonstrated the potential of grape pomace, a major winemaking by-product, to be upcycled as a functional ingredient for plant-based yogurt alternatives. First, an extensive characterization of the gurts fermented with selected starters was performed before and after fermentation, as well as during refrigerated storage, leading to the selection of the gurt fermented with *Lp. plantarum* T0A10.

Then, an in-depth focus on the effect of fermentation on phenolic compounds and their related antioxidant efficacy on human keratinocytes was provided, elucidating a compound/function relationship. These findings provide a foundation for developing value-added products from grape pomace, supporting both circular economy initiatives and functional ingredient innovation. Future work should incorporate consumer-based sensory testing to validate perceptual trends identified by the trained panel. Moreover, given the heterogeneity of plant-derived matrices, the behavior observed in this study could be confirmed on a larger scale with GP deriving from different winemaking processes.

## Figures and Tables

**Figure 1 foods-14-04294-f001:**
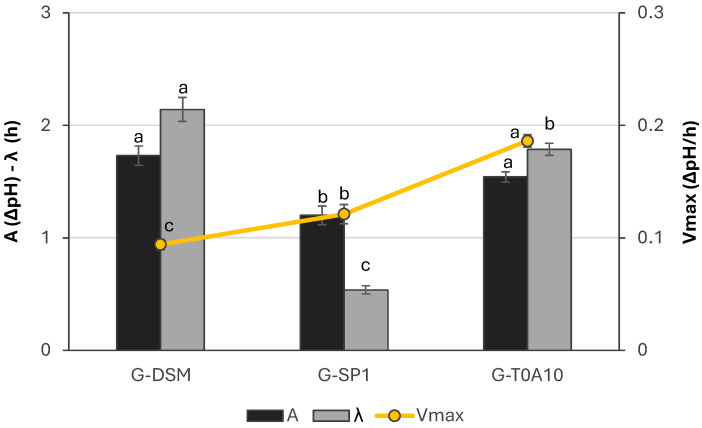
Acidification kinetics parameters of the gurts containing rice and grape pomace fermented with *Leuc. pseudomesenteroides* DMS20193 (G-DSM), *Lc. rhamnosus* SP1 (G-SP1), and *Lp. plantarum* T0A10 (G-T0A10). ^a–c^ Values with different letters differ significantly (*p* < 0.05).

**Figure 2 foods-14-04294-f002:**
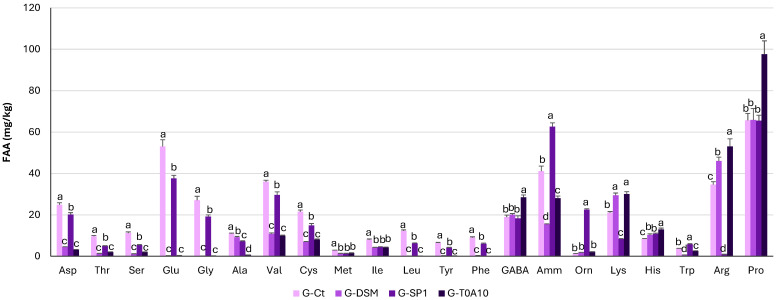
Free amino acids concentration expressed as mg/kg of plant-based gurts. G-Ct—control gurt; G-DSM, G-SP1, and G-T0A10—gurts fermented with *Leuc. pseudomesenteroides* DMS20193, *Lc. rhamnosus* SP1, and *Lp. plantarum* T0A10, respectively. ^a–c^ Values with different letters differ significantly (*p* < 0.05).

**Figure 3 foods-14-04294-f003:**
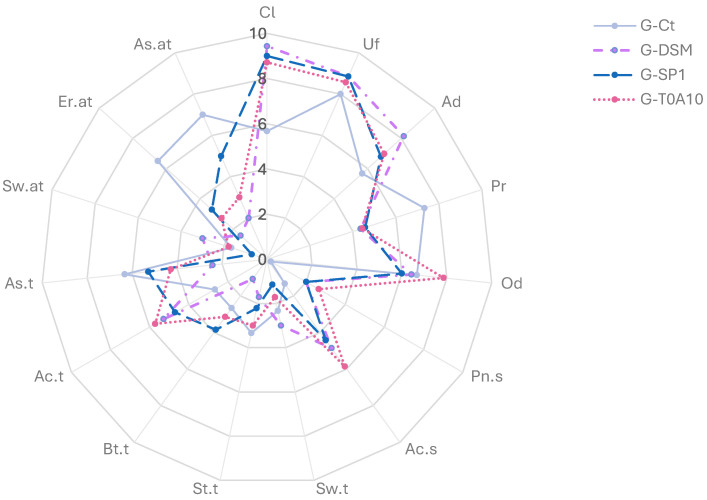
Sensory analysis of plant-based gurts. G-Ct—control gurt; G-DSM, G-SP1, and G-T0A10—gurts fermented with *Leuc. pseudomesenteroides* DMS20193, *Lc. rhamnosus* SP1, and *Lp. plantarum* T0A10, respectively.

**Figure 4 foods-14-04294-f004:**
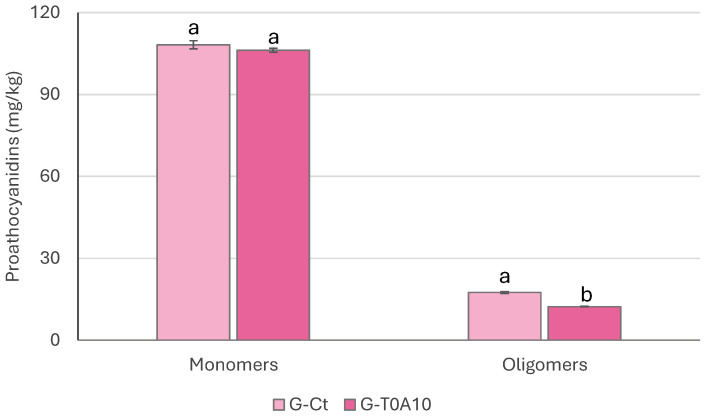
Proanthocyanidin content (mg/kg) in plant-based gurts. G-Ct—control gurt; G-T0A10—gurt fermented with *Lp. plantarum* T0A10. ^a,b^ Values with different letters differ significantly (*p* < 0.05).

**Figure 5 foods-14-04294-f005:**
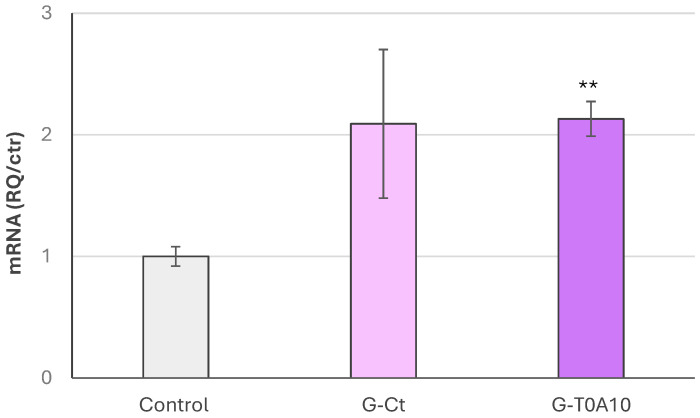
Gene expression analysis of *SOD2* determined by qRT-PCR of NCTC 2544 cell line treated for 24 h with plant-based gurts supplemented with grape pomace before (G-Ct) and after fermentation with *Lp. plantarum* T0A10 (G-T0A10). Columns represent average value, and error bars represent SEM. Asterisks indicate significant difference compared to control cells (** *p* < 0.01).

**Table 1 foods-14-04294-t001:** Microbiological and biochemical characterization of plant-based gurts at the end of fermentation (tf) and after 7 (t7) and 14 (t14) days of refrigerated storage. G-Ct—control gurt; G-DSM, G-SP1, and G-T0A10—gurts fermented with *Leuc. pseudomesenteroides* DMS20193, *Lc. rhamnosus* SP1, and *Lp. plantarum* T0A10, respectively.

	LAB (cfu/g)	pH	TTA (mL)	Lactic Acid (mmol/kg)	Acetic Acid (mmol/kg)	TFAA (mg/kg)	TPC (mmol GAE/kg)	DPPH (%)
tf								
G-Ct	2.70 ± 0.12 ^Bb^	6.07 ± 0.09 ^Aa^	1.8 ± 0.03 ^Bc^	0.66 ± 0.02 ^Bc^	0.15 ± 0.01 ^Bc^	427 ± 20 ^Aa^	756 ± 3.63 ^Ac^	40.1 ± 2.26 ^Ac^
G-DSM	8.96 ± 0.41 ^Aa^	4.37 ± 0.03 ^Ab^	4.6 ± 0.14 ^Ba^	14.80 ± 0.58 ^Ba^	0.69 ± 0.02 ^Bb^	269 ± 12 ^Bc^	828 ± 2.24 ^Aa^	58.8 ± 1.44 ^Aa^
G-SP1	8.56 ± 0.25 ^Aa^	4.79 ± 0.10 ^Ab^	3.4 ± 0.23 ^Cb^	11.49 ± 0.61 ^Bb^	0.59 ± 0.03 ^Bb^	394 ± 14 ^Ba^	803 ± 0.13 ^Ab^	50.5 ± 1.21 ^Bab^
G-T0 A10	8.84 ± 0.38 ^Aa^	4.49 ± 0.05 ^Ab^	4.2 ± 0.17 ^Ba^	14.66 ± 0.43 ^Ba^	1.35 ± 0.07 ^Ba^	305 ± 5 ^Cb^	839 ± 1.09 ^Ba^	56.9 ± 2.93 ^Ca^
t7								
G-Ct	2.71 ± 0.09 ^Bc^	5.97 ± 0.11 ^Aa^	2.0 ± 0.12 ^Ac^	0.72 ± 0.01 ^Ac^	0.23 ± 0.02 ^Ac^	456 ± 23 ^Aa^	750 ± 1.52 ^Ac^	38.3 ± 3.57 ^Ad^
G-DSM	8.95 ± 0.31 ^Aa^	4.23 ± 0.10 ^Ac^	5.4 ± 0.11 ^Aa^	15.98 ± 0.75 ^Aa^	0.71 ± 0.09 ^Ab^	294 ± 18 ^ABc^	832 ± 1.04 ^ABa^	62.2 ± 1.34 ^Ab^
G-SP1	8.29 ± 0.14 ^Ab^	4.53 ± 0.03 ^Bb^	3.7 ± 0.07 ^Bb^	12.91 ± 0.34 ^Ab^	0.81 ± 0.05 ^Ab^	408 ± 33 ^ABab^	807 ± 2.93 ^Ab^	53.7 ± 3.29 ^Bc^
G-T0A10	9.07 ± 0.42 ^Aa^	4.34 ± 0.09 ^Ac^	5.2 ± 0.14 ^Aa^	15.17 ± 0.29 ^Aa^	1.91 ± 0.21 ^Aa^	358 ± 12 ^Bb^	847 ± 4.14 ^Ba^	69.2 ± 1.23 ^Ba^
t14								
G-Ct	3.06 ± 0.07 ^Ab^	5.78 ± 0.05 ^Ba^	2.3 ± 0.09 ^Ad^	0.81 ± 0.03 ^Ac^	0.32 ± 0.01 ^Ac^	437 ± 9 ^Aa^	755 ± 0.25 ^Ac^	36.3 ± 2.16 ^Ac^
G-DSM	8.85 ± 0.11 ^Ba^	3.98 ± 0.07 ^Bd^	6.0 ± 0.37 ^Aa^	16.22 ± 0.11 ^Aa^	0.73 ± 0.10 ^Ab^	325 ± 3 ^Ab^	838 ± 2.24 ^Aa^	65.3 ± 1.89 ^Ab^
G-SP1	8.18 ± 0.72 ^Aa^	4.56 ± 0.17 ^Bb^	4.1 ± 0.19 ^Ac^	13.45 ± 0.61 ^Ab^	0.84 ± 0.05 ^Ab^	451 ± 14 ^Aa^	799 ± 2.79 ^Ab^	67.2 ± 1.07 ^Ab^
G-T0A10	8.92 ± 0.49 ^Aa^	4.23 ± 0.04 ^Bc^	5.2 ± 0.13 ^Ab^	15.36 ± 0.37 ^Ab^	1.89 ± 0.17 ^Aa^	427 ± 20 ^Aa^	887 ± 0.08 ^Aa^	77.9 ± 2.58 ^Aa^

Values in the same column with different uppercase letters, among different storage times of the same sample, denote significant differences at *p* < 0.05. Different lowercase letters, among different samples within the same storage time, denote significant differences at *p* < 0.05.

**Table 2 foods-14-04294-t002:** Chromaticity coordinates of plant-based gurts before (G-Ct) and after fermentation with *Leuc. pseudomesenteroides* DMS20193, *Lc. rhamnosus* SP1, and *Lp. plantarum* T0A10 (G-DSM, G-SP1 and G-T0A10, respectively).

	*L**	*a**	*b**	Δ*E**
G-Ct	42.5 ± 0.9 ^a^	−0.42 ± 0.09 ^c^	1.82 ± 0.22 ^a^	42.4 ± 1.01 ^a^
G-DSM	42.0 ± 0.31 ^a^	1.30 ± 0.23 ^ab^	1.91 ± 0.12 ^a^	42.5 ± 0.29 ^a^
G-SP1	42.7 ± 0.4 ^a^	0.41 ± 0.22 ^b^	1.86 ± 0.13 ^a^	42.8 ± 0.31 ^a^
G-T0A10	41.4 ± 0.3 ^a^	2.19 ± 0.49 ^a^	1.97 ± 0.13 ^a^	42.2 ± 0.51 ^a^

Values in the same column with different superscript letters differ significantly at *p* < 0.05.

**Table 3 foods-14-04294-t003:** Phenolic compounds (expressed as mg/kg) identified in plant-based gurts supplemented with grape pomace before (G-Ct) and after fermentation with *Lp. plantarum* T0A10 (G-T0A10).

Phenolic Compound	G-Ct	G-T0A10
Gallic acid	14.1 ± 2.7 ^a^	6.0 ± 0.3 ^b^
Pyrogallol	n.d.	4.4 ± 0.8 ^a^
Syringol	0.9 ± 0.1 ^a^	n.d.
Protocatechuic acid	0.7 ± 0.1 ^a^	n.d.
Protocatechuic acid isomer II	8.2 ± 1.3 ^a^	4.0 ± 0.1 ^b^
Caftaric acid	1.9 ± 0.4 ^a^	n.d.
Methyl gallate	3.3 ± 0.5 ^a^	3.9 ± 0.1 ^a^
Hydroxybenzoic acid	11.1 ± 2.3 ^a^	9.9 ± 3.9 ^a^
Hydroxybenzoic acid isomer II	1.0 ± 0.1 ^a^	n.d.
Proanthocyanidin B1	8.8 ± 0.2 ^b^	14.3 ± 1.4 ^a^
Caffeoyl glucose	7.1 ± 0.9 ^a^	5.0 ± 0.6 ^a^
(+)-Catechin	70.9 ± 0.9 ^b^	99.3 ± 8.8 ^a^
Caffeoyl glucose isomer	5.2 ± 0.6 ^a^	5.0 ± 0.0 ^a^
Fertaric acid	2.3 ± 0.5 ^a^	n.d.
Proanthocyanidin B1 isomer II	1.8 ± 0.5 ^a^	n.d.
*p*-Coumaric acid glucoside	12.2 ± 2.9 ^b^	16.2 ± 0.4 ^a^
Proanthocyanidin B1 isomer III	10.3 ± 0.4 ^b^	12.1 ± 0.6 ^a^
Syringic acid	6.3 ± 0.7 ^a^	3.7 ± 0.2 ^b^
Protocatechuic acid isomer III	4.7 ± 0.1 ^b^	8.8 ± 1.6 ^a^
*p*-Coumaric acid glucoside isomer II	13.6 ± 4.3 ^a^	11.8 ± 6.2 ^a^
(−)-epicatechin isomer II	45.0 ± 1.5 ^b^	59.7 ± 0.6 ^a^
*p*-Coumaric acid	1.6 ± 0.1 ^a^	n.d.
Syringic acid isomer II	15.3 ± 2.4 ^a^	10.4 ± 0.2 ^b^
Proanthocyanidin B1 gallate	3.0 ± 0.0 ^b^	4.4 ± 1.3 ^a^
Proanthocyanidin B1 isomer IV	0.3 ± 0.2 ^a^	n.d.
Myricetin glucuronide	1.8 ± 0.5 ^a^	1.5 ± 0.2 ^a^
*p*-Coumaric acid isomer II	1.8 ± 0.5 ^a^	1.3 ± 0.4 ^a^
Hydroxybenzoic acid isomer II	3.2 ± 0.1 ^a^	3.0 ± 0.3 ^a^
Malvidin glucoside pyruvic acid	0.7 ± 0.1 ^a^	1.1 ± 0.3 ^a^
Epicatechin gallate	1.6 ± 0.0 ^b^	3.4 ± 0.5 ^a^
Ellagic acid	1.2 ± 0.7 ^a^	0.9 ± 0.4 ^a^
Isoquercitrin	3.0 ± 0.6 ^a^	2.5 ± 0.5 ^a^
Quercetin glucuronide	46.2 ± 1.9 ^a^	36.5 ± 2.3 ^b^
Isoquercitrin isomer II	6.5 ± 1.0 ^a^	5.1 ± 0.8 ^a^
Eriodictyol	0.6 ± 0.1 ^a^	0.8 ± 0.1 ^a^
Isorhamnetin galactoside	0.5 ± 0.0 ^a^	n.d.
Isorhamnetin galactoside isomer II	9.0 ± 1.0 ^a^	7.2 ± 0.9 ^a^
Syringetin glucoside	18.0 ± 1.5 ^a^	16.7 ± 1.8 ^a^
Quercetin	2.4 ± 0.0 ^a^	5.7 ± 1.2 ^b^

^a,b^ Values with different superscript letters differ significantly (*p* < 0.05). n.d.—not detected.

## Data Availability

The contributions presented in this study are included in the article and Appendix A. Further inquiries can be directed to the corresponding author.

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
