# Peer review of "Upcycling Grape Pomace in a Plant-Based Yogurt Alternative: Starter Selection, Phenolic Profiling, and Antioxidant Efficacy on Human Keratinocytes"

_foods, 2025, doi:10.3390/foods14244294_

Round 1
Reviewer 1 Report
Comments and Suggestions for Authors
I read the manuscript 'Upcycling Grape Pomace in a Plant-based Yogurt Alternative: Starter selection, Phenolic Profiling, and Antioxidant Efficacy on Human Keratinocytes', investigates how fermenting a rice-based plant gurt enriched with grape pomace alters its phenolic composition and enhances its antioxidant effects, ultimately improving its ability to protect human keratinocytes against oxidative stress.
The research is well planned. Very detailed description of measurement methods and very detailed discussion of results.
In my opinion, the abstract is a bit too long, not very synthetic, and lacks the presentation of numerical analysis results.
Lines 157-159 - detoxify/detoxified it's not the best stylistically or terminologically. In the scientific literature on DPPH, the standard words used are “scavenge”, “reduce”, and so on.
2.5.2 - used two times.
Lines 371-372 - I think that the differences presented in the table should be discussed in more detail.
Line 438 - "extreme acidic" - I don't see the pH of GP indicated anywhere in the content.
I really appreciate articles that present photos of the products obtained, I feel that this is a bit lacking here.
I'm also missing a color assessment, especially since you mention the product's color in several places. I assume the GP in your study contains colorants, especially in the case of red grapes, which was your variety (Vitis vinifera L. cultivar Primitivo).
I also miss an explanation of why the Human keratinocyte NCTC 2544 cell line (HL97002) was chosen. It would be logical and not require comment if the product being analyzed was for skin use (cosmetic), but in the case of a food product, solid justification is required.
I think the article is well-written, and the research is important because it shows that a by-product of the wine industry — grape pomace — can be transformed into a functional food ingredient, supporting a circular economy. It's also interesting because it reveals how fermentation changes the structure of phenolic compounds and increases their biological activity, significantly improving the protection of human skin cells from oxidative stress.
Author Response
I read the manuscript 'Upcycling Grape Pomace in a Plant-based Yogurt Alternative: Starter selection, Phenolic Profiling, and Antioxidant Efficacy on Human Keratinocytes', investigates how fermenting a rice-based plant gurt enriched with grape pomace alters its phenolic composition and enhances its antioxidant effects, ultimately improving its ability to protect human keratinocytes against oxidative stress.
The research is well planned. Very detailed description of measurement methods and very detailed discussion of results.
The authors thank the reviewer for the comment. All suggestion were taken into consideration, and a point-by-point revision was provided.
In my opinion, the abstract is a bit too long, not very synthetic, and lacks the presentation of numerical analysis results.
Ok. The comment was taken into consideration and the abstract modified including more numerical results.
Lines 157-159 - detoxify/detoxified it's not the best stylistically or terminologically. In the scientific literature on DPPH, the standard words used are “scavenge”, “reduce”, and so on.
Ok. The correct term was used
2.5.2 - used two times.
Ok
Lines 371-372 - I think that the differences presented in the table should be discussed in more detail.
Ok. Differences in phenolic compounds were highlighted in more detail in the results and discussion section (Lines 389-393, 516-526)
Line 438 - "extreme acidic" - I don't see the pH of GP indicated anywhere in the content.
The reviewer is correct, that comment referred to previous literature on the subject, a reference was added in support of the statement
I really appreciate articles that present photos of the products obtained, I feel that this is a bit lacking here. I'm also missing a color assessment, especially since you mention the product's color in several places. I assume the GP in your study contains colorants, especially in the case of red grapes, which was your variety (Vitis vinifera L. cultivar Primitivo).
The reviewer is correct GP conferred a purple color to the gurt; hence the color analysis was performed and chromaticity coordinates for each gurt included in the manuscript (lines 365-371).
I also miss an explanation of why the Human keratinocyte NCTC 2544 cell line (HL97002) was chosen. It would be logical and not require comment if the product being analyzed was for skin use (cosmetic), but in the case of a food product, solid justification is required.
Thank you for the comment. We agree that, since the product under study is a food matrix, the rationale for using the NCTC 2544 human keratinocyte cell line must be clarified. Indeed, our aim was to investigate the biological relevance of the antioxidant potential of grape pomace–based gurts using a well-established human cell model responsive to oxidative stress. Keratinocytes are among the most widely used human cell lines for screening antioxidant and anti-inflammatory bioactivity of plant bioactive compounds. Although the product is not intended for topical use and further studies could focus on evaluating gastrointestinal effects of the fermented gurt designed in this study, phenolic compounds absorbed from foods can exert systemic antioxidant and cytoprotective effects, including in peripheral tissues such as skin.
This aspect was deepened in the discussion (lines 534-543), clarifying the rationale for using this cell line and emphasizing that the in vitro model was selected for mechanistic assessment rather than to represent the product’s intended route of consumption.
I think the article is well-written, and the research is important because it shows that a by-product of the wine industry — grape pomace — can be transformed into a functional food ingredient, supporting a circular economy. It's also interesting because it reveals how fermentation changes the structure of phenolic compounds and increases their biological activity, significantly improving the protection of human skin cells from oxidative stress.
The authors thank the reviewer for the comment
Reviewer 2 Report
Comments and Suggestions for Authors
Torreggiani et al. have aimed to design a rice/grape pomace-yogurt alternative (Gurt) fermented with selected lactic acid bacteria, optimize starter selection, analyze phenolics and evaluate antioxidant efficiency on human keratinocytes. This is an interesting piece of work and should be of interest in the field. However, the following issues need to addressed for a possible publication in Foods:
Abstract
- Provide abstract in one single paragraph. Several long sentences in abstract should be split for easy understanding. Also, vague terms like ‘good pro-technological performances’ should be made specific.
- Too much methodological detail (like glycosylated forms and pyrogallol formation) should be replaced with focused outcomes. Also, claims of ‘significant up-regulation’ should be suffixed with p-values (p<0.05).
Keywords
- Some more keywords such as ‘polyphenol biotransformation’, ‘Lactiplantibacillus plantarum’ and ‘gurt fermentation’; should be added.
Introduction
- Using phrases like ‘much of the state-of-the-art research’ should be supported by citations.
- Claims on environmental impacts and consumer trends should be provided with precise references.
- Consider splitting long and information-dense sentences to improve readability.
Materials and methods
- GP drying conditions (65°C for 60 min) seems insufficient unless pre-dried.
- Inoculum preparation lacks details of washing steps, solvent use (tap water is not acceptable).
- Polyphenol and amino acid analyses lack extraction ratios and internal standard details.
- Concentration of gentamycin (50 mg/mL) and hydrogen peroxide (50 mM) are too high as the respective typical values should be 50 microgram/mL and 0.1-0.5 mM.
- Student’s test is not ‘non-parametric’ and should be corrected.
- Quantitation details lack calibration curves and R2 values.
- Sensory evaluation lacks an ethics approval statement.
Results
- Section 4.1 – R2 should be reported for Gompertz model. Also, biomass comparison in L296-298 should include statistics.
- Section 4.2 – ‘slightly increased proteins’ needs numerical input. Also, statistical significance for all compositional data should be clearly reported.
- Section 4.3 – TTA expression should be clarified, that is, “g lactic acid/100 g ?). Also, link between acidity and sensory outcomes should be provided to improve cohesion.
- Section 4.4 – Increase in amino acid should be quantified and discussed for nutritional relevance.
- Section 4.5 – increases/decreases should be quantified in mg/g or percentage. Also, visual integration using figures/tables is required.
Discussion
- The discussion should address the limitation such as (1) high variability in plant-based matrices, (2) potential off-flavors from GP, (3) high hydrogen peroxide levels and (4) sensory panel short comings.
- Some more critical interpretation is required instead of just restating the results. Also careful reorganization to reduce redundancy is a must.
Conclusion
- A better distinguishing between demonstrated outcomes versus potential applications is important. Also, future work suggestions would strengthen the final statements (for example, scaling, shelf-life optimization and consumer studies).
Author Response
Torreggiani et al. have aimed to design a rice/grape pomace-yogurt alternative (Gurt) fermented with selected lactic acid bacteria, optimize starter selection, analyze phenolics and evaluate antioxidant efficiency on human keratinocytes. This is an interesting piece of work and should be of interest in the field.
The authors thank the reviewer for the comment. All suggestion were taken into consideration, and a point-by-point revision was provided.
However, the following issues need to addressed for a possible publication in Foods:
Abstract
- Provide abstract in one single paragraph. Several long sentences in abstract should be split for easy understanding. Also, vague terms like ‘good pro-technological performances’ should be made specific.
- Too much methodological detail (like glycosylated forms and pyrogallol formation) should be replaced with focused outcomes. Also, claims of ‘significant up-regulation’ should be suffixed with p-values (p<0.05).
The abstract was revised according to suggestions.
Keywords
- Some more keywords such as ‘polyphenol biotransformation’, ‘Lactiplantibacillus plantarum’ and ‘gurt fermentation’; should be added.
As suggested new keywords were added.
Introduction
- Using phrases like ‘much of the state-of-the-art research’ should be supported by citations.
Ok. The statement was revised for clarity and supported by relevant references.
- Claims on environmental impacts and consumer trends should be provided with precise references.
Ok. The statement was supported by relevant references.
- Consider splitting long and information-dense sentences to improve readability.
As suggested the text was revised to improve readability.
Materials and methods
- GP drying conditions (65°C for 60 min) seems insufficient unless pre-dried.
For the drying process, small amount of GP were evenly placed in perforated trays to speed up the process and limit oxidation.
- Inoculum preparation lacks details of washing steps, solvent use (tap water is not acceptable).
Details on inoculum preparation are reported in lines 113-115. As for the solvent, any solution commonly used to wash the cells (Ringer solution, phosphate buffer, physiological solution) would have interfered with the organoleptic properties of the gurts since they tend to confer bitter or salty notes. Hence tap water was used to resuspend the cells right before the inoculum, ensuring that both the vitality of the cells and the sensory properties of the gurts were not affected.
- Polyphenol and amino acid analyses lack extraction ratios and internal standard details.
Ok. More details on phenolic ancids and amino acids extraction were added (lines 146-148; 198-200).
- Concentration of gentamycin (50 mg/mL) and hydrogen peroxide (50 mM) are too high as the respective typical values should be 50 microgram/mL and 0.1-0.5 mM.
We agree that the concentrations reported seemed unusually high, however, those value refer to the stock solution added to the culture medium, not to the final concentration in the wells. The final concentration used in cell culture was 50 µg/mL for gentamycin and 0.5 mM for hydrogen peroxyde, fully in line with standard values. We have now corrected the text to clearly specify the final working concentration.
- Student’s test is not ‘non-parametric’ and should be corrected.
Ok. Welch's correction was performed to the t-test to adjusts the degrees of freedom and to accurately compare the means of two groups, making it more robust and reliable than the standard Student's t-test.
- Quantitation details lack calibration curves and R2 values.
Ok. We appreciate the remark. We have now expanded the analytical section to include description of the calibration curves used for quantification of phenolic compounds, the concentration ranges covered for each standard, R² values of the regression equations.
- Sensory evaluation lacks an ethics approval statement.
All necessary information regarding the ethical statement were provided
Results
- Section 4.2 – ‘slightly increased proteins’ needs numerical input. Also, statistical significance for all compositional data should be clearly reported.
A section 4.2 is not existent in the manuscript, nor it is the statement ‘slightly increased proteins’.
- Section 4.3 – TTA expression should be clarified, that is, “g lactic acid/100 g ?). Also, link between acidity and sensory outcomes should be provided to improve cohesion.
TTA expression unit (quantity of 0.1 M NaOH needed to reach a pH of 8.3 expressed as mL) was clearly reported in the materials and method section (lines 142-144) and the units reported after each TTA result. A link between acidity and sensory notes was added to the discussion (lines 494-499).
- Section 4.4 – Increase in amino acid should be quantified and discussed for nutritional relevance.
A section 4.4 is not existent in the manuscript, also, most amino acids decreased in fermented samples, this aspect was abundantly discussed.
- Section 4.5 – increases/decreases should be quantified in mg/g or percentage. Also, visual integration using figures/tables is required.
A section 4.5 is not existent in the manuscript.
Discussion
- The discussion should address the limitation such as (1) high variability in plant-based matrices, (2) potential off-flavors from GP, (3) high hydrogen peroxide levels and (4) sensory panel short comings.
- Some more critical interpretation is required instead of just restating the results. Also careful reorganization to reduce redundancy is a must.
As suggested all these aspects were taken into consideration in revising the discussion.
Conclusion
- A better distinguishing between demonstrated outcomes versus potential applications is important. Also, future work suggestions would strengthen the final statements (for example, scaling, shelf-life optimization and consumer studies).
As suggested all these aspects were taken into consideration in revising the conclusion.
Reviewer 3 Report
Comments and Suggestions for Authors
In this study, a fermented plant-based gurt enriched with grape pomace was described through a two-stage investigation: screening LAB-fermented plant-based gurt prototypes containing grape pomace based on suitable biochemical, nutritional, and functional quality; and selecting the most promising beverage prototypes for in-depth phenolic profiling and antioxidant testing in human keratinocytes. The article is highly relevant and makes a valuable contribution to the field of food waste valorization and functional food development.
Suggestions for improvements:
Page 3, line 103, please provide proximate carbohydrate content for the used grape pomace
Page 4, line 191, please explain the selection of the mixture of ethanol/water 80/20 (v/v) and provide a reference.
Page 4, please provide the Institutional Review Board document number for the Sensory analysis.
Page 9, Figure 3: Please consider using different colours to improve the visibility of the sensory analysis illustration.
Pages 12-14: since the Discussion section is well elaborated and comprehensive, please consider separating the Conclusions from the Discussion to highlight the main findings further.
Author Response
In this study, a fermented plant-based gurt enriched with grape pomace was described through a two-stage investigation: screening LAB-fermented plant-based gurt prototypes containing grape pomace based on suitable biochemical, nutritional, and functional quality; and selecting the most promising beverage prototypes for in-depth phenolic profiling and antioxidant testing in human keratinocytes. The article is highly relevant and makes a valuable contribution to the field of food waste valorization and functional food development.
The authors thank the reviewer for the comment. All suggestion were taken into consideration, and a point-by-point revision was provided.
Suggestions for improvements:
Page 3, line 103, please provide proximate carbohydrate content for the used grape pomace
Carbohydrate content was provided.
Page 4, line 191, please explain the selection of the mixture of ethanol/water 80/20 (v/v) and provide a reference.
Overall, hydro-alcoholic mixtures like ethanol/water usually extract the broadest range of grape phenolics. Moreover, although mildly acidified hydro-alcoholic mixtures are preferred, many optimization studies on grape pomace report either optimal performance near 50–70% ethanol or use 70–80% ethanol with excellent yields; comparable to 80% methanol. Thus 80% ethanol was selected for this study. A reference was added in the text.
Page 4, please provide the Institutional Review Board document number for the Sensory analysis.
All necessary information regarding the ethical statement were provided
Page 9, Figure 3: Please consider using different colours to improve the visibility of the sensory analysis illustration.
As suggested colours were modified to improve readability
Pages 12-14: since the Discussion section is well elaborated and comprehensive, please consider separating the Conclusions from the Discussion to highlight the main findings further.
As suggested the conclusion was separated from the discussion section.
Round 2
Reviewer 2 Report
Comments and Suggestions for Authors
The authors have satisfactorily addressed all the comments raised by reviewers and substantially improved the overall quality of the article.